# Dramatic resurgence of malaria after 7 years of intensive vector control interventions in Eastern Uganda

**Moses R. Kamya**[1,2]*, **Joaniter I. Nankabirwa**[1,2], **Emmanuel Arinaitwe**[1], **John Rek**[1,3], **Maato Zedi**[1], **Catherine Maiteki-Sebuguzi**[1,3], **Jimmy Opigo**[3], **Sarah G. Staedke**[4], **Ambrose Oruni**[4], **Martin J. Donnelly**[4], **Bryan Greenhouse**[5], **Jessica Briggs**[5], **Paul J. Krezanoski**[5], **Teun Bousema**[6], **Philip J. Rosenthal**[5], **Peter Olwoch**[1], **Prasanna Jagannathan**[7], **Isabel Rodriguez-Barraquer**[5], **Grant Dorsey**[5]

**1** Infectious Diseases Research Collaboration, Kampala, Uganda, **2** School of Medicine, Makerere University Kampala, Kampala, Uganda, **3** National Malaria Control Division, Ministry of Health, Kampala, Uganda, **4** Department of Vector Biology, Liverpool School of Tropical Medicine, Liverpool, United Kingdom, **5** Department of Medicine, University of California San Francisco, San Francisco, California, United States of America, **6** Department of Medical Microbiology, Radboud University Medical Center, Nijmegen, Netherlands, **7** Division of Infectious Diseases and Geographic Medicine, Department of Medicine, Stanford University, Stanford, California, United States of America

* mkamya@idrc-uganda.org

**Data Availability Statement:** The dataset underlying this study is available in the ClinEpiDB database (DS_17191d35b9).

## Abstract

Tororo District, Uganda experienced a dramatic decrease in malaria burden from 2015–19 during 5 years of indoor residual spraying (IRS) with carbamate (Bendiocarb) and then organophosphate (Actellic) insecticides. However, a marked resurgence occurred in 2020, which coincided with a change to a clothianidin-based IRS formulations (Fludora Fusion/SumiShield). To quantify the magnitude of the resurgence, investigate causes, and evaluate the impact of a shift back to IRS with Actellic in 2023, we assessed changes in malaria metrics in regions within and near Tororo District. Malaria surveillance data from Nagongera Health Center, Tororo District was included from 2011–2023. In addition, a cohort of 667 residents from 84 houses was followed from August 2020 through September 2023 from an area bordering Tororo and neighboring Busia District, where IRS has never been implemented. Cohort participants underwent passive surveillance for clinical malaria and active surveillance for parasitemia every 28 days. Mosquitoes were collected in cohort households every 2 weeks using CDC light traps. Female *Anopheles* were speciated and tested for sporozoites and phenotypic insecticide resistance. Temporal comparisons of malaria metrics were stratified by geographic regions. At Nagongera Health Center average monthly malaria cases varied from 419 prior to implementation of IRS; to 56 after 5 years of IRS with Bendiocarb and Actellic; to 1591 after the change in IRS to Fludora Fusion/SumiShield; to 155 after a change back to Actellic. Among cohort participants living away from the border in Tororo, malaria incidence increased over 8-fold (0.36 vs. 2.97 episodes per person year, p<0.0001) and parasite prevalence increased over 4-fold (17% vs. 70%, p<0.0001) from 2021 to 2022 when Fludora Fusion/SumiShield was used. Incidence decreased almost 5-fold (2.97 vs. 0.70, p<0.0001) and prevalence decreased by 39% (70% vs. 43%, p<0.0001) after shifting back to Actellic. There was a similar pattern among those living near the border in Tororo,

**Funding:** This project was funded by the US National Institutes of Health as part of the International Centers of Excellence in Malaria Research (ICEMR) program (U19AI089674 to GD). The funders played no role in the design of the study; in the collection, analyses, and interpretation of data; in the writing of the manuscript; or in the decision to submit the manuscript for publication.

**Competing interests:** The authors have declared that no competing interests exist.

with increased incidence between 2021 and 2022 (0.93 vs. 2.40, p<0.0001) followed by a decrease after the change to Actellic (2.40 vs. 1.33, p<0.001). Among residents of Busia, malaria incidence did not change significantly over the 3 years of observation. Malaria resurgence in Tororo was temporally correlated with the replacement of *An. gambiae s.s.* by *An. funestus* as the primary vector, with a marked decrease in the density of *An. funestus* following the shift back to IRS with Actellic. In Busia, *An. gambiae s.s.* remained the primary vector throughout the observation period. Sporozoite rates were approximately 50% higher among *An. funestus* compared to the other common malaria vectors. Insecticide resistance phenotyping of *An. funestus* revealed high tolerance to clothianidin, but full susceptibility to Actellic. A dramatic resurgence of malaria in Tororo was temporally associated with a change to clothianidin-based IRS formulations and emergence of *An. funestus* as the predominant vector. Malaria decreased after a shift back to IRS with Actellic. This study highlights the ability of malaria vectors to rapidly circumvent control efforts and the importance of high-quality surveillance systems to assess the impact of malaria control interventions and generate timely, actionable data.

## Introduction

Between 2000 and 2015, it was estimated the incidence of malaria in Sub-Saharan Africa decreased by 40%, with the scale up of vector control interventions responsible for the majority of cases averted [1]. However, since 2015 progress has stalled and even reversed course in some of the highest burden countries of Africa [2]. Indeed, the global burden of malaria has become increasingly concentrated in Africa, which accounted for over 94% of cases and 96% of deaths in 2022 [2]. Turning the tide on malaria will require a better understanding of the root causes of stalled progress, better use of local data to inform policy decision making, and a more flexible and targeted approach to control interventions [3].

Uganda is emblematic of other high burden African countries, where progress in reducing the burden of malaria has been slow and difficult to sustain despite the scale up of proven control interventions. Uganda was the first country to implement universal distribution of free long-lasting insecticidal nets (LLINs) starting in 2014, with repeated campaigns every 3–4 years. Uganda also has one of the largest indoor residual spraying of insecticide (IRS) programs, focusing on selected high transmission districts using different formulations of insecticides rotated every few years. The success of this intensive approach to vector control has been well documented in Tororo District, a historically high transmission area of southeastern Uganda, where our group has been conducting comprehensive cohort-based malaria surveillance studies since 2011 [4]. IRS was first implemented in Tororo District in December 2014, initially using a carbamate (Bendiocarb), then switching to an organophosphate (Actellic) in 2016. Comparing key malaria indicators prior to IRS (2011–14) and after two rounds of universal LLIN distribution and 5 years of sustained IRS (2017–19), we documented a 500-fold decrease in malaria transmission intensity, a 60-fold decrease in the incidence of symptomatic malaria, and a 5-fold decrease in parasite prevalence among children 0.5–10 years of age [5]. In addition, there was a marked shift in the predominant vector species from *An. gambiae s.s.* to *An. arabiensis* between these two time periods [5]. However, following a change to clothianidin-based formulations of IRS in 2019–20, a marked resurgence of malaria cases was

documented using health facility-based data from 5 districts of Uganda (including Tororo), reaching pre-IRS levels within 1–2 years [6].

To better quantify the magnitude of the malaria resurgence in Tororo and to investigate potential causes, we compared temporal changes in malaria incidence, prevalence, and entomological measures between September 2020 and September 2023 in a cohort of 667 residents living in two areas within Tororo District and in neighboring Busia District, where IRS has never been implemented. In addition, we assessed changes after the formulation of IRS in Tororo District was shifted back to an organophosphate in March 2023.

## Methods

### Study setting and population level vector control interventions

Our team has conducted a series of comprehensive cohort-based malaria surveillance studies (referred to as PRISM) in southeastern Uganda starting in October 2011 (Fig 1). This area is characterized by perennial transmission with two seasonal peaks following the rainy seasons. The "PRISM 1" study was conducted from October 2011 –September 2017 and the "PRISM 2" study conducted from October 2017 –October 2019 in Nagongera subcounty, Tororo District [5]. The "PRISM Border Cohort" study (the focus of this report) was conducted from August 2020 –September 2023 in both Tororo District and neighboring Busia District. Tororo is historically a high malaria transmission district with an estimated entomological inoculation rate (EIR) of 310 infective bites per person per year in 2011–12 [7]. Prior to 2013, vector control in Tororo District was limited to the distribution of LLINs through antenatal care services. In November 2013, universal distribution of free LLINs was conducted as part of a national campaign, and similar campaigns were repeated in May 2017 and June 2020. All LLIN distribution

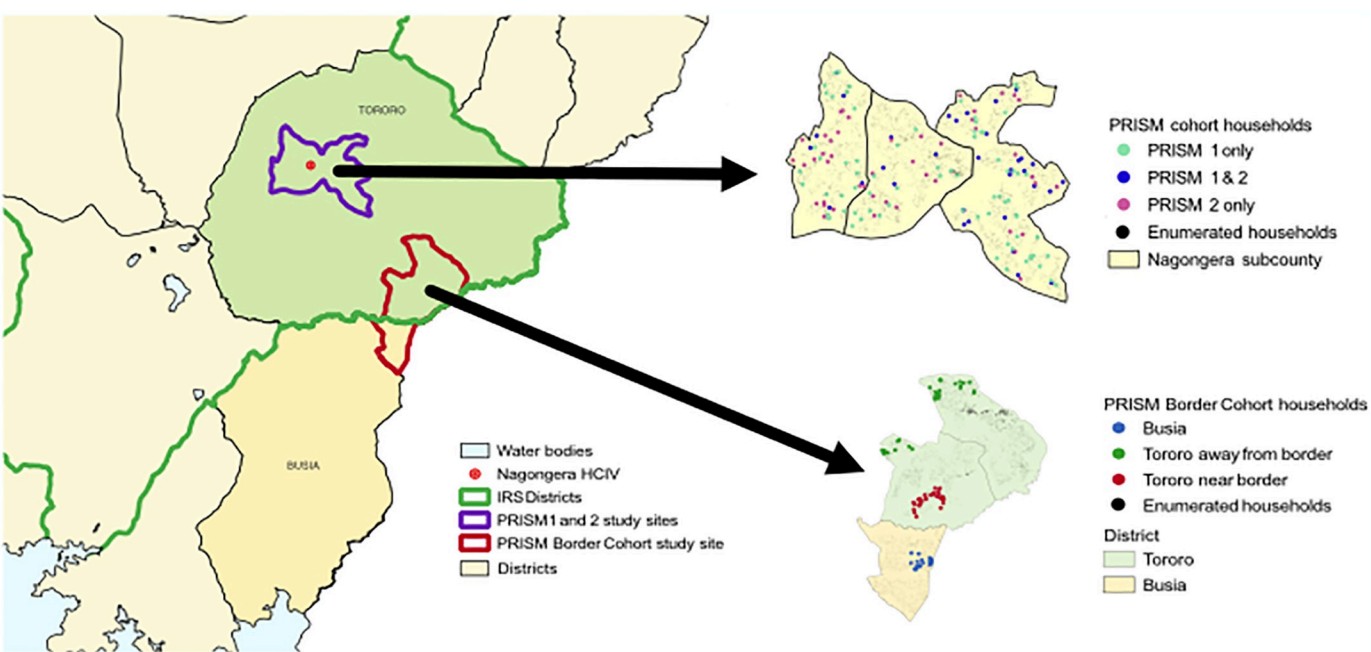

**Fig 1. Map of eastern Uganda with an extract showing study sites.** This map was created using shapefiles of the districts, parishes, and villages of Uganda obtained from the Humanitarian Data Exchange v1.80.0 PY3, linked here: https://data.humdata.org/dataset/uganda-administrative-boundaries-as-of-17-08-2018. Explicit data licensing terms can be found here: https://data.humdata.org/faqs/licenses.

campaigns in Tororo District utilized standard pyrethroid LLINs. IRS with the carbamate (Bendiocarb) was introduced for the first time in Tororo in December 2014–January 2015, with additional rounds administered in June–July 2015 and November–December 2015. In June–July 2016, the IRS compound was changed to an organophosphate (Actellic), with repeated rounds in June–July 2017, June–July 2018, and March–April 2019 [5]. The IRS program changed to clothianidin-based formulations in 2020, with Fludora Fusion (clothianidin + deltamethrin) administered in March 2020 and March 2021, and SumiShield (clothianidin alone) administered in March 2022. In March 2023, a decision was made to shift back to IRS with Actellic in response to the resurgence of malaria. Busia is a high malaria transmission district bordering Tororo District to the south. As in Tororo prior to 2013, vector control in Busia District was limited to targeted distribution of LLINs through antenatal care services. Universal distribution campaigns of free LLINs were conducted in Busia District in May 2013, May 2017, and December 2020. The first two LLINs distribution campaigns in Busia District utilized standard pyrethroid LLINs and the third campaign utilized LLINs containing deltamethrin plus piperonyl butoxide (PermaNet 3.0). IRS has never been implemented in Busia District.

### Routine health facility-based malaria surveillance data

To provide context, we illustrate continuous trends in disease burden from October 2011 – September 2023 at Nagongera Health Center IV, Tororo District (Fig 2). This health center is part of a national health facility-based malaria surveillance network that provides high-quality data to monitor geographic and temporal trends in malaria burden and assess the impact of population level control interventions [6]. Routine individual level data are collected from all patients who present to the outpatient departments, including the number of laboratory confirmed cases of malaria. Since November 2018, after mapping of the region, we have been able to estimate the incidence of malaria from a target area around Nagongera Health Center IV, defined as the number of cases of laboratory-confirmed malaria diagnosed at the health center

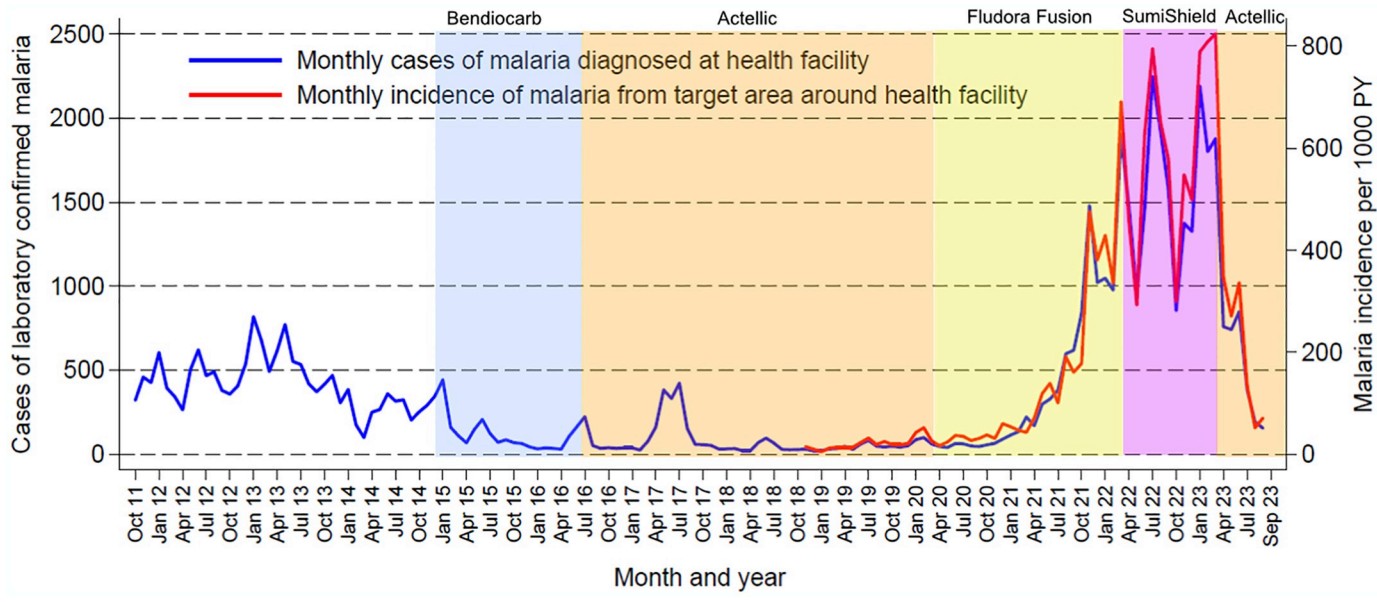

**Fig 2. Monthly trends in laboratory confirmed cases of malaria and malaria incidence from Nagongera Health Center.** PY = person years. Color schemes indicate periods during which different formulations of IRS were implemented including periods of expected residual activity.

among patients residing in the target area, per unit time, divided by the total population of the target area.

## Screening and enrollment of households into the cohort study

Screening and enrollment of households and study participants was previously described [8]. In May 2020, all households in a contiguous study area including two sub-counties in Tororo District and one sub-county in Busia District were enumerated and mapped using handheld global positioning systems (Garmin e-Trex 10 GPS unit, Garmin International Inc., Olathe, KS) to provide a sampling frame for recruitment of households into the cohort study. Of note, there were no houses located along the border between the two districts due to the presence of a river and adjacent swampy areas. Households were enrolled from three distinct geographic regions (Fig 1). In August 2020, 20 households from Busia District located 1.9–3.2 km from the border with Tororo District and 30 households from Tororo district located 0.7–3.5 km from the border (referred to as Tororo, near border) were enrolled. Preliminary data revealed that malaria burden was surprisingly similar between houses in Busia District and those in Tororo near the border [8]. To capture more of a gradient in transmission intensity, an additional 30 households from Tororo District located 5.5–10.8 km from the border (referred to as Tororo, away from border) were enrolled in January 2021. Households were randomly selected from our enumeration list for screening and enrolled if they met the following criteria: 1) having at least two members aged 5 years or younger; 2) no more than 7 permanent residents currently residing; 3) no plans to move from the study catchment area in the next 2 years; and 4) willingness to participate in entomological surveillance studies. In March 2022, 4 households (3 from Busia and 1 from Tororo away from the border) were enrolled to replace households where all participants had been withdrawn from the cohort study.

## Screening, enrollment and follow-up of cohort participants

All permanent residents from enrolled households were screened and enrolled in the cohort study if they met the following criteria: 1) the selected household was considered their primary residence; 2) agreement to come to the study clinic for any febrile illness and scheduled routine visits; 3) agreement to avoid antimalarial medications outside the study; and 4) provision of written informed consent (for parent or guardian in the case of children). The cohort was dynamic, such that over the course of the study any permanent residents that joined a household were screened for enrollment. At enrollment, a baseline evaluation was conducted including a detailed medical history, focused physical examination, and blood collection by venipuncture for hemoglobin measurement, thick blood smear, and storage for future molecular studies. A household survey was conducted to collect information on characteristics of the household and LLIN ownership; all household members were provided access to an LLIN after the survey. A wealth index was generated for each household using principal components analysis based on common assets and categorized into tertiles. Cohort study participants were encouraged to come to a dedicated study clinic open 7 days per week for all their medical care. Routine visits were conducted every 4 weeks and included a standardized evaluation and collection of blood by finger prick/heel stick (if < 6 months of age) or venipuncture (if aged 6 months and older) for thick blood smear, hemoglobin measurement (every 12 weeks), and storage for future molecular studies. Study participants found to have a fever (tympanic temperature > 38.0°C) or history of fever in the previous 24 hours at the time of any clinic visit had a thick blood smear read immediately. If the thick blood smear was positive by light microscopy, the patient was diagnosed with malaria and managed according to national guidelines. Study participants who missed their scheduled routine visits were visited at home and

requested to come to the study clinic as soon as possible. All enrolled participants were followed through September 30th, 2023 unless they were prematurely withdrawn. Participants were withdrawn if they: 1) moved out of the cohort household; 2) were unable to be located for > 4 months; 3) withdrew informed consent; or 4) were unable to comply with the study schedule and procedures.

## Laboratory evaluations

Thick blood smears were stained with 2% Giemsa for 30 minutes and evaluated for the presence of asexual parasites. Parasite densities were calculated by counting the number of asexual parasites per 200 leukocytes (or per 500, if the count was less than 10 parasites per 200 leukocytes), assuming a leukocyte count of 8,000/μL. A thick blood smear was considered negative if examination of 100 high power fields revealed no asexual parasites. For quality control, all slides were read by a second microscopist, and a third reviewer settled any discrepant readings. Quantitative PCR (qPCR) was performed at the time of enrollment, at each routine visit (every 4 weeks), and when malaria was diagnosed. At each of these time points, DNA was extracted from approximately 200 μL of whole blood using Qiagen spin columns, and extraction products were tested for the presence and quantity of *P. falciparum* DNA using a highly sensitive qPCR assay targeting the multicopy conserved var gene acidic terminal sequence with a lower limit of detection of 50 parasite/mL [9].

## Entomological surveillance

Mosquito collections were conducted every 2 weeks in all rooms of study houses where cohort study participants slept using CDC light traps positioned 1 m above the floor (Model 512; John W. Hock Company, Gainesville, Florida, USA). Traps were set at 7 PM and contents collected at 7 AM the following morning. Entomology technicians assessed whether cohort study participants reported sleeping under an LLIN each morning when trap contents were collected. All female *Anopheles* were enumerated and identified taxonomically to species level based on morphological criteria according to established taxonomic keys [10]. Every 2 weeks, up to 30 mosquitoes identified as from the *An. gambiae s.l.* complex from each of the 3 study sites were randomly selected for PCR analysis to distinguish *An. gambiae s.s.* from *An. arabiensis* [11]. All female Anopheles mosquitoes were stored in desiccant and up to 50 mosquitoes per CDC light trap collection were assessed for sporozoites using ELISA [12].

## Insecticide resistance phenotyping

Insecticide resistance testing was performed using field caught mosquitoes collected from neighboring Mayuge district as part of a separate study to compare the response of both *An. funestus s.l.* and *An. gambiae s.l.* to clothianidin and pirimiphos-methyl. Blood fed indoor-resting mosquitoes were collected in May 2023 using a Prokopack electric aspirator, kept for 3–4 days, and then made to lay eggs by forced-egg laying [13]. The F1 progeny were reared and then used for bioassays using CDC bottles for clothianidin and WHO tubes for pirimiphos-methyl at the standard diagnostic dose as per WHO protocol [14]. Mosquitoes were exposed at different times to assess the time response curve for both species and then kept for 5 days post exposure to clothianidin and 1 day post exposure to pirimiphos-methyl to assess delayed mortality.

## Data analyses

All data were collected using standardized case record forms and double-entered using Microsoft Access (Microsoft Corporation, Redmond, Washington, USA). Analyses were performed using Stata, version 14 (Stata Corporation, College Station, Texas, USA). The primary objective of the study was to compare temporal trends in key malaria metrics in the context of population level vector control interventions stratified by 3 geographic regions: 1) Tororo, away from border, 2) Tororo, near border, and 3) Busia (Fig 1). The incidence of symptomatic malaria was defined as the number of incident cases of malaria (fever plus a positive thick blood smear by microscopy) divided by the person time of observation. Episodes of malaria occurring within 10 days of a prior episode were not considered incident events. The prevalence of microscopic parasitemia was defined as the proportion of routine visits conducted every 4 weeks in which asexual parasites were detected by microscopy. When estimating the prevalence of microscopic or sub-microscopic parasitemia, samples that were positive by qPCR but negative by microscopy were added to the numerator. Vector density was defined as the number of female *Anopheles* collected (stratified by species) divided by the number CDC light trap collections. The number of *An. gambiae s.s.* and *An. arabiensis* collected were estimated by multiplying the observed proportions using the subset tested with species PCR by the total number of *An. gambiae s.l.* collected stratified by each 2-week collection period and geographic region. Selected comparisons of measures of malaria incidence were made using mixed effects generalized linear models with a negative binomial regression family, with person time of observation included as an offset, and random effects at the level of the individual. Selected comparisons of measures of parasitemia at the time of routine visits were made using mixed effects generalized linear models with a Poisson family and random effects at the level of the individual. Selected comparisons of measures of vector density were made using negative binomial regression models with the number of CDC light trap collections included as an offset. For all comparisons, a p-value < 0.05 was considered statistically significant.

## Ethics approval and informed consent

Ethical approval was obtained from the Makerere University School of Medicine Research and Ethics Committee (REF 2019–134), the Uganda National Council for Science and Technology (HS 2700), the London School of Hygiene & Tropical Medicine Ethics Committee (17777), and the University of California, San Francisco Committee on Human Research (257790). Written informed consent was obtained for all participants prior to enrolment into the cohort study.

## Results

### Temporal changes in malaria burden from Nagongera Health Center

Data on temporal changes in malaria burden from Nagongera Health Center (located in the interior of Tororo District) covering from approximately 3 years before until 9 years after the initiation of IRS are presented in Fig 2. From October 2011 –November 2014, prior to the implementation of IRS, an average of 419 laboratory confirmed cases were diagnosed each month. After IRS was implemented, first with Bendiocarb (administered every 6 months) followed by Actellic (administered annually), there was a sharp decline in malaria burden, reaching an average of 56 cases per month and an incidence of 25 episodes per 1000 person years in the surrounding community from March 2019 –February 2020. After switching to annual rounds of IRS with clothianidin-based formulations in March 2020, malaria burden gradually began to increase. This increase accelerated after the 2nd round of Fludora Fusion in March

2021. From March 2022 –February 2023, following a round of SumiShield (the 3[rd] and final round of clothianidin-based IRS) malaria burden reached an average of 1591 cases per month and an incidence of 587 episodes per 1000 person years. Malaria burden declined sharply after switching back to IRS with Actellic in March 2023, reaching 155 cases and an incidence of 71 episodes per 1000 person years in September 2023.

## Characteristics of cohort households and study participants

A cohort study including 3 distinct geographic regions in Tororo and Busia districts was designed to further characterize the resurgence of malaria and explore potential causes. Characteristics of the households and cohort participants at the time of enrollment are presented in Table 1. Household characteristics unrelated to vector control interventions were similar across the three geographic regions, with the exception of household wealth, as a higher proportion of houses in Busia were in the poorest category compared to houses in Tororo. The proportion of houses that reported owning at least one LLIN at enrollment ranged from 30% in Busia to 74% in Tororo, away from border. Adequate LLIN coverage (defined as 1 LLIN per 2 persons) was 19% and similarly low across the three geographic regions. All cohort households were provided LLINs by the study team following enrollment and 90% of household residents reported sleeping under an LLIN at the time of home assessments conducted every 2 weeks over the course of follow-up, with no significant differences between the three geographic regions. As expected, no households in Busia reported receiving IRS 12 months prior to enrollment or during the course of the study. Household coverage for the 4 annual rounds of IRS conducted in Tororo district from 2020 (assessed at enrollment) through 2023 was 90–100% within Tororo, near border and 80–100% within Tororo, away from border.

**Table 1. Characteristics of households and cohort participants at enrolment.**

| Characteristic | | Tororo District | | Busia District |
|---|---|---|---|---|
| | | Away from border | Near border | |
| **Household characteristics** | | | | |
| Number of Households | | 31 | 30 | 23 |
| Residents per household, median (range) | | 7 (4–7) | 7 (4–7) | 5 (3–7) |
| Type of housing construction, n (%) | Traditional | 20 (64.5) | 18 (60.0) | 13 (56.5) |
| | Modern | 11 (35.5) | 12 (40.0) | 10 (43.5) |
| Wealth category, n (%) | Poorest | 5 (16.1) | 7 (23.3) | 13 (56.5) |
| | Middle | 15 (48.4) | 10 (33.3) | 4 (17.4) |
| | Least poor | 11 (35.5) | 13 (43.3) | 6 (26.1) |
| Number of rooms used for sleeping, median (range) | | 2 (1–3) | 2 (1–4) | 1 (1–3) |
| Number of sleeping spaces, median (range) | | 3 (2–4) | 3 (2–4) | 2 (2–6) |
| IRS in the last 12 months, n (%) | | 29 (93.6) | 30 (100) | 0 |
| Households with at least 1 LLIN, n (%) | | 23 (74.2) | 16 (53.3) | 7 (30.4) |
| Households with 1 LLIN per 2 persons, n (%) | | 7 (22.6) | 5 (16.7) | 4 (17.4) |
| **Participant characteristics** | | | | |
| Number of participants | | 246 | 235 | 186 |
| Female gender, n (%) | | 146 (59.4) | 125 (53.2) | 98 (52.7) |
| Age in years, median (IQR) | | 11.1 (3.8–24.2) | 9.2 (3.8–22.4) | 8.5 (3.2–22.3) |
| Age categories, n (%) | < 5 years | 83 (33.7) | 81 (34.5) | 68 (36.6) |
| | 5–15 years | 76 (30.9) | 82 (34.9) | 60 (32.3) |
| | > 15 years | 87 (35.4) | 72 (30.6) | 58 (31.2) |

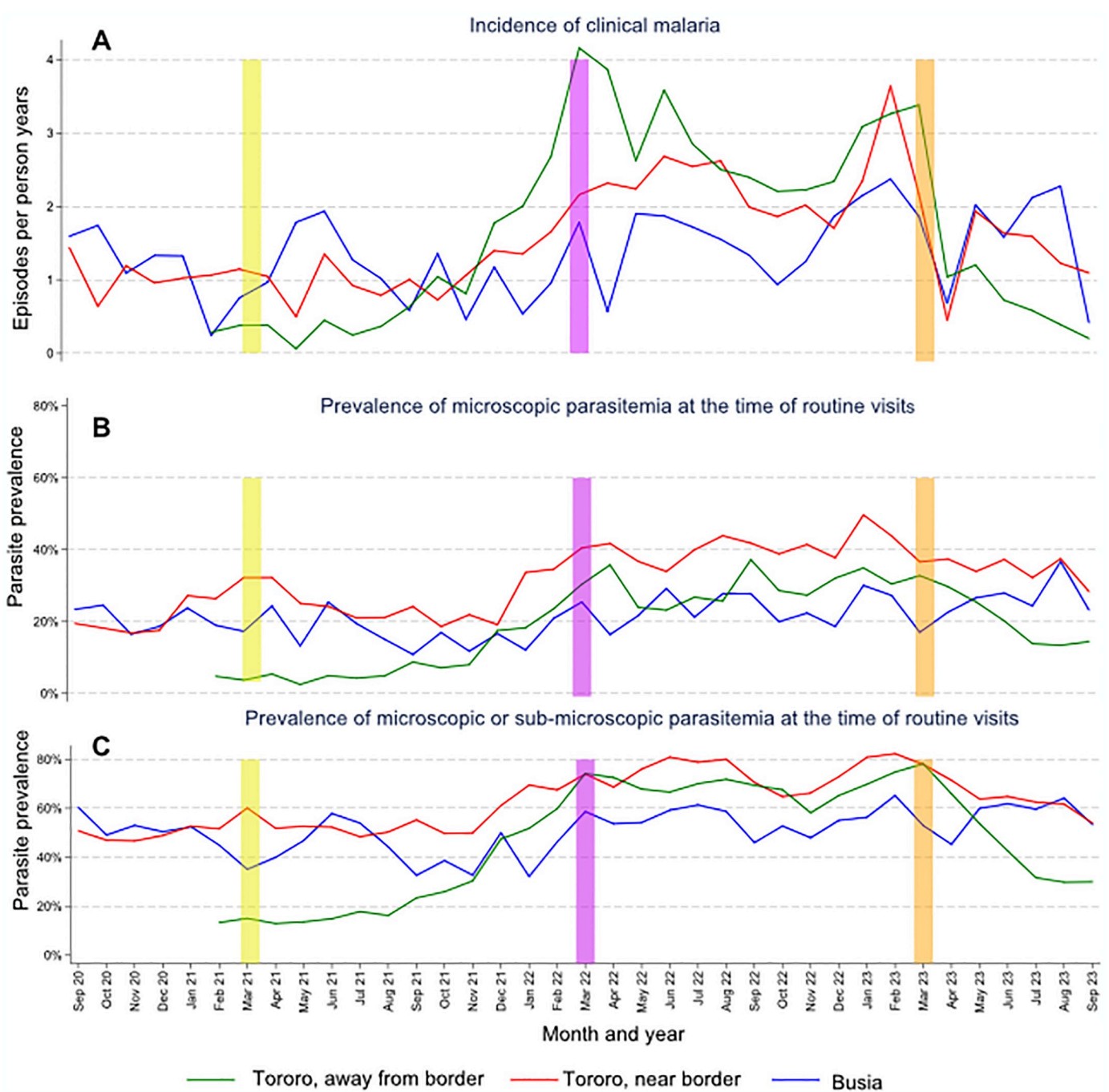

**Fig 3. Monthly trends in the incidence of clinical malaria, prevalence of microscopic parasitemia, and prevalence of microscopic or sub-microscopic parasitemia stratified by geographic regions.** Vertical bars indicate rounds of IRS (Tororo District only): yellow = Fludora Fusion, purple = SumiShield, and orange = Actellic.

### Temporal changes in the incidence of malaria

Monthly trends over the entire observation period in the incidence of malaria among cohort participants of all ages are presented in Fig 3A stratified by geographic region. Among residents of all ages from Tororo, away from border, the incidence of clinical malaria remained

below 0.5 episodes per person year (PY) for the first 7 months of observation from February-August 2021 before there was a sharp increase, reaching over 4 episodes per PY in March 2022. The incidence remained above 2.2 episodes per PY for all months from April 2022 until Actellic was sprayed in March 2023, after which there was a sharp decline, reaching 0.2 episodes per PY in September 2023 (Fig 3A). To allow for direct comparisons over time and in relationship to rounds of IRS, malaria incidence (stratified by geographic region and age categories) was quantified over 6-month time periods from April 2021 through September 2023 (Table 2). Among residents from Tororo, away from border, the incidence of malaria increased over 8-fold between 2021 and 2022 (0.36 vs. 2.97, p<0.0001) corresponding to the second and third rounds of IRS with clothianidin-based formulations, then decreased almost 5-fold between 2022 and 2023 (2.97 vs. 0.70, p<0.0001) following the shift back to Actellic. Among residents from Tororo, near border, the incidence of clinical malaria increased over 2.5-fold between April-September of 2021 and 2022 (0.93 vs. 2.40, p<0.0001), then decreased almost 2-fold between 2022 and 2023 (2.40 vs. 1.33, p<0.0001). Among residents of Busia, the incidence of malaria between April-September did not change significantly from 2021 to 2022 to 2023. Across the 3 geographic regions, the incidence of malaria was between 3 and 4 times higher in children compared to adults, and the temporal patterns described above were similar across different age categories (Table 2). Despite the high burden of malaria in this study, only 4 of 2,266 episodes (0.2%) met WHO criteria for severe malaria: 2 cases of severe anemia (Hb < 5.0 gm/dL) occurred in children with confirmed sickle cell disease, one of whom died; 1 case of severe anemia occurred in a 1 year old child who recovered; and 1 case of jaundice occurred in a 3 year old child who recovered.

## Temporal changes in parasite prevalence

Temporal changes in parasite prevalence mirrored those seen for incidence. Among residents of all ages from Tororo, away from border, the prevalence of microscopic parasitemia remained below 6% and the prevalence of microscopic or sub-microscopic parasitemia remained below 18% over the first 7 months of observation (February-August 2021) before there was a sharp increase, reaching over 34% and 78%, respectively, in the year after the final round of clothianidin-based IRS was administered (Fig 3B and 3C). After IRS was changed back to Actellic in March 2023, there was a sharp decline in these measures, with parasitemia reaching 13% and 30%, respectively. Comparing 6-month time periods from April–September following annual rounds of IRS, the prevalence of microscopic parasitemia increased over 5-fold (5% vs. 29%, p<0.0001) and the prevalence of microscopic or sub-microscopic parasitemia increased over 4-fold (17% vs 70%, p<0.0001) between April-September of 2021 and April-September of 2022. After spraying with Actellic, the prevalence of microscopic parasitemia decreased by 30% (29% vs. 20%, p<0.0001) and the prevalence of microscopic or sub-microscopic parasitemia decreased by 39% (70% vs 43%, p<0.0001). Among residents from Tororo, near border, the prevalence of microscopic parasitemia increased by 62% (24% vs. 40%, p<0.0001) and the prevalence of microscopic or sub-microscopic parasitemia increased by 47% (52% vs. 76%, p<0.0001) between April-September of 2021 and April-September of 2022. After spraying with Actellic, the prevalence of microscopic parasitemia decreased by 13% (40% vs. 35%, p = 0.04) and the prevalence of microscopic or sub-microscopic parasitemia decreased by 17% (76% vs. 63%, p<0.0001). Among residents from Busia, between April-September of 2021 and April-September of 2022 there were modest increases in the prevalence of microscopic parasitemia (18% vs. 24%, p = 0.03) and microscopic or sub-microscopic parasitemia (46% vs. 56%, p = 0.03), and, in contrast to the areas in Tororo receiving IRS with Actellic, remained similarly high in April-September of 2023. Across the 3 geographic regions,

**Table 2. Temporal changes in measures of malaria burden stratified by age categories and study site.**

| Study site | Age category | 6-month time periods following the 2nd round of Fludora Fusion in Tororo district | | | | |
|---|---|---|---|---|---|---|
| | | Apr 21 –Sep 21 | Oct 21 –Mar 22 | Apr 22 –Sep 22 | Oct 22 –Mar 23 | Apr 23 –Sep 23 |
| **Incidence of clinical malaria = number of cases / person years of observation (episodes per person year)** | | | | | | |
| Tororo, away from border | All ages | 34/95.5 (0.36) | 197/95.2 (2.07) | 289/97.3 (2.97) | 260/94.8 (2.74) | 64/91.9 (0.70) |
| | < 5 years | 15/30.8 (0.49) | 83/28.4 (2.92) | 114/28.4 (4.01) | 106/25.5 (4.16) | 13/20.2 (0.64) |
| | 5–15 years | 12/27.9 (0.43) | 81/30.7 (2.64) | 140/35.5 (3.94) | 124/35.7 (3.47) | 34/36.5 (0.93) |
| | > 15 years | 7/36.7 (0.19) | 33/36.1 (0.91) | 35/33.3 (1.05) | 30/33.6 (0.89) | 17/35.2 (0.48) |
| Tororo, near border | All ages | 89/95.5 (0.93) | 132/95.4 (1.38) | 219/91.3 (2.40) | 216/95.3 (2.27) | 123/92.6 (1.33) |
| | < 5 years | 43/28.2 (1.53) | 61/23.1 (2.64) | 73/18.8 (3.89) | 67/16.9 (3.96) | 24/14.0 (1.72) |
| | 5–15 years | 38/35.2 (1.08) | 58/40.2 (1.44) | 116/42.3 (2.75) | 123/47.6 (2.58) | 78/47.4 (1.65) |
| | > 15 years | 8/32.2 (0.25) | 13/32.1 (0.40) | 30/30.3 (0.99) | 26/30.7 (0.85) | 21/31.3 (0.67) |
| Busia | All ages | 70/54.9 (1.27) | 59/55.2 (1.07) | 97/64.8 (1.50) | 109/62.9 (1.73) | 93/60.8 (1.53) |
| | < 5 years | 30/16.6 (1.81) | 18/15.1 (1.19) | 30/15.8 (1.89) | 30/13.5 (2.21) | 21/11.5 (1.83) |
| | 5–15 years | 31/19.3 (1.61) | 33/22.2 (1.49) | 56/28.5 (1.96) | 65/30.0 (2.17) | 58/30.6 (1.90) |
| | > 15 years | 9/19.1 (0.47) | 8/17.9 (0.45) | 11/20.4 (0.54) | 14/19.4 (0.72) | 14/18.7 (0.75) |
| **Prevalence of microscopic parasitemia at the time of routine visits performed every 4 weeks** | | | | | | |
| Tororo, away from border | All ages | 61/1177 (5.2%) | 213/1255 (17.0%) | 346/1209 (28.6%) | 384/1250 (30.7%) | 227/1145 (19.8%) |
| | < 5 years | 9/374 (2.4%) | 60/380 (15.8%) | 97/351 (27.6%) | 88/343 (25.7%) | 29/255 (11.4%) |
| | 5–15 years | 33/341 (9.7%) | 91/399 (22.8%) | 157/444 (35.4%) | 188/475 (39.6%) | 143/451 (31.7%) |
| | > 15 years | 19/462 (4.1%) | 62/476 (13.0%) | 92/414 (22.2%) | 108/432 (25.0%) | 55/439 (12.5%) |
| Tororo, near border | All ages | 323/1325 (24.4%) | 318/1139 (27.9%) | 503/1266 (39.7%) | 472/1143 (41.3%) | 436/1259 (34.6%) |
| | < 5 years | 63/387 (16.3%) | 62/275 (22.5%) | 98/262 (37.4%) | 74/205 (36.1%) | 49/188 (26.1%) |
| | 5–15 years | 203/492 (41.3%) | 200/483 (41.4%) | 307/587 (52.3%) | 312/574 (54.4%) | 291/654 (44.5%) |
| | > 15 years | 57/446 (12.8%) | 56/381 (14.7%) | 98/417 (23.5%) | 86/364 (23.6%) | 96/417 (23.0%) |
| Busia | All ages | 121/662 (18.3%) | 145/794 (18.3%) | 185/772 (24.0%) | 195/858 (22.7%) | 195/721 (27.0%) |
| | < 5 years | 28/198 (14.1%) | 22/221 (10.0%) | 32/188 (17.0%) | 32/187 (17.1%) | 29/137 (21.2%) |
| | 5–15 years | 67/235 (28.5%) | 94/319 (29.5%) | 119/345 (34.5%) | 117/406 (28.8%) | 120/359 (33.4%) |
| | > 15 years | 26/229 (11.4%) | 29/254 (11.4%) | 34/239 (14.2%) | 46/265 (17.4%) | 46/225 (20.4%) |
| **Prevalence of microscopic or sub-microscopic parasitemia at the time of routine visits performed every 4 weeks** | | | | | | |
| Tororo, away from border | All ages | 195/1177 (16.6%) | 593/1255 (47.3%) | 843/1209 (69.7%) | 853/1250 (68.2%) | 487/1145 (42.5%) |
| | < 5 years | 42/374 (11.2%) | 158/380 (41.6%) | 238/351 (67.8%) | 232/343 (67.6%) | 79/255 (31.0%) |
| | 5–15 years | 79/341 (23.2%) | 229/399 (57.4%) | 339/444 (76.4%) | 335/475 (70.5%) | 215/451 (47.7%) |
| | > 15 years | 74/462 (16.0%) | 206/476 (43.3%) | 266/414 (64.3%) | 286/432 (66.2%) | 193/439 (44.0%) |
| Tororo, near border | All ages | 687/1325 (51.8%) | 704/1139 (61.8%) | 966/1266 (76.3%) | 847/1143 (74.1%) | 795/1259 (63.1%) |
| | < 5 years | 141/387 (36.4%) | 160/275 (58.2%) | 200/262 (76.3%) | 155/205 (75.6%) | 99/188 (52.7%) |
| | 5–15 years | 329/492 (66.9%) | 335/483 (69.4%) | 487/587 (83.0%) | 481/574 (83.8%) | 464/654 (70.9%) |
| | > 15 years | 217/446 (48.7%) | 209/381 (54.9%) | 279/417 (66.9%) | 211/364 (58.0%) | 232/417 (55.6%) |
| Busia | All ages | 306/662 (46.2%) | 356/794 (44.8%) | 429/772 (55.6%) | 478/858 (55.7%) | 413/721 (57.3%) |
| | < 5 years | 76/198 (38.4%) | 73/221 (33.0%) | 77/188 (41.0%) | 81/187 (43.3%) | 64/137 (46.7%) |
| | 5–15 years | 125/235 (53.2%) | 180/319 (56.4%) | 217/345 (62.9%) | 249/406 (61.3%) | 224/359 (62.4%) |
| | > 15 years | 105/229 (45.9%) | 103/254 (40.6%) | 135/239 (56.5%) | 148/265 (55.8%) | 125/225 (55.6%) |

the prevalence of parasitemia was highest among children 5–15 years of age and the temporal patterns described above were similar across different age categories.

## Temporal changes in entomological measures

Among houses from Tororo, away from border, through December 2021 the predominant species was *An. arabiensis* (71% of female *Anopheles* collected) followed by *An. funestus* (14%)

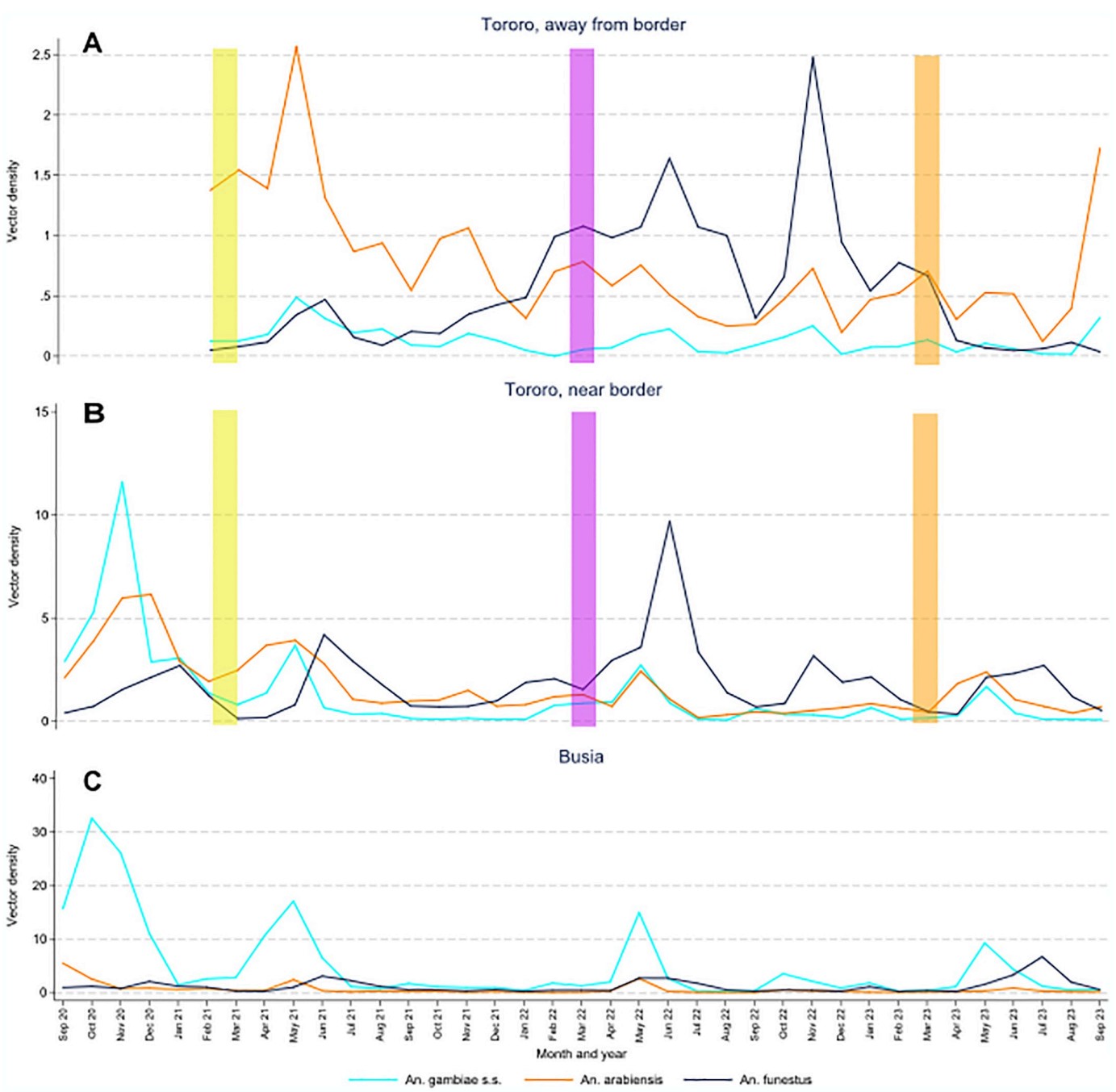

**Fig 4. Monthly trends in the vector densities of common *Anopheles* species stratified by geographic regions.** Note the scale of the y-axes differ by geographic regions. Vertical bars indicate rounds of IRS: yellow = Fludora Fusion, purple = SumiShield, and orange = Actellic.

and *An. gambiae s.s.* (12%). After December 2021, *An. funestus* became the predominant species, a trend that continued until after IRS was shifted back to Actellic in March 2023, when the density of *An. funestus* decreased sharply (Fig 4A). Between mid-2021 and mid-2022, the density of *An. funestus* increased over 4-fold (0.2 vs 1.0 mosquitoes per collection, p<0.0001); this density then decreased over 10-fold (1.0 vs 0.08, p<0.0001) between mid-2022 and mid-2023. During these same 6-month time periods, the densities of *An. gambiae s.s.* and *An.*

**Table 3. Temporal changes in vector density stratified by Anopheles species and study site.**

| Study site | Anopheles species | 6-month time periods following the 2nd round of Fludora Fusion in Tororo district | | | | | | | | | | Sporozoite rate (full study period) |
| | | Apr 21 –Sep 21 | | Oct 21 –Mar 22 | | Apr 22 –Sep 22 | | Oct 22 –Mar 23 | | Apr 23 –Sep 23 | | |
| | | N[a] | n[b] (VD)[c] | N[a] | n[b] (VD)[c] | N[a] | n[b] (VD)[c] | N[a] | n[b] (VD)[c] | N[a] | n[b] (VD)[c] | |
| Tororo, away from border | *An. gambiae s.s.* | 739 | 179 (0.2) | 713 | 61 (0.09) | 732 | 80 (0.1) | 728 | 88 (0.1) | 740 | 66 (0.09) | 4/484 (0.83%) |
| | *An. arabiensis* | | 919 (1.2) | | 518 (0.7) | | 327 (0.4) | | 379 (0.5) | | 434 (0.6) | 31/2818 (1.10%) |
| | *An. funestus* | | 170 (0.2) | | 419 (0.6) | | 749 (1.0) | | 757 (1.0) | | 56 (0.08) | 43/2160 (1.99%) |
| | Other anopheles | | 54 (0.07) | | 32 (0.04) | | 18 (0.02) | | 21 (0.03) | | 21 (0.03) | 1/155 (0.65%) |
| Tororo, near border | *An. gambiae s.s.* | 750 | 769 (1.0) | 763 | 276 (0.4) | 744 | 604 (0.8) | 749 | 222 (0.3) | 747 | 298 (0.4) | 35/5312 (0.66%) |
| | *An. arabiensis* | | 1577 (2.1) | | 832 (1.1) | | 604 (0.8) | | 446 (0.6) | | 842 (1.1) | 86/7194 (1.20%) |
| | *An. funestus* | | 1313 (1.8) | | 1024 (1.3) | | 2570 (3.5) | | 1196 (1.6) | | 1154 (1.5) | 125/7978 (1.57%) |
| | Other anopheles | | 60 (0.08) | | 23 (0.03) | | 80 (0.1) | | 28 (0.04) | | 153 (0.2) | 3/482 (0.62%) |
| Busia | *An. gambiae s.s.* | 397 | 2657 (6.7) | 395 | 421 (1.1) | 472 | 2042 (4.3) | 463 | 728 (1.6) | 472 | 1535 (3.3) | 143/10500 (1.36%) |
| | *An. arabiensis* | | 259 (0.7) | | 27 (0.07) | | 320 (0.7) | | 92 (0.2) | | 128 (0.3) | 13/1196 (1.09%) |
| | *An. funestus* | | 563 (1.4) | | 145 (0.4) | | 694 (1.5) | | 207 (0.4) | | 1107 (2.3) | 62/2982 (2.08%) |
| | Other anopheles | | 19 (0.05) | | 5 (0.01) | | 16 (0.03) | | 12 (0.03) | | 27 (0.06) | 2/114 (1.75%) |

[a] N = number of CDC LT collections,

[b] n = number of female anopheles mosquitoes collected

[c] VD = vector density (n/N)

*arabiensis* decreased between 2021 and 2022, with no significant change from 2022 to 2023 (Table 3). Among houses from Tororo, near border, vector densities were initially more evenly distributed across the 3 major *Anopheles* species. After December 2021, *An. funestus* became the predominant species until after the round of IRS with Actellic in March 2023, although the decrease in *An. funestus* was not as sustained as in houses away from the border (Fig 4B and Table 3). In Busia, *An. gambiae s.s.* made up 73% of all female *Anopheles* collected and was the predominant species throughout the observation period, with the exception of June-August each year, when *An. funestus* was predominant (Fig 4C and Table 3).

Overall, sporozoite rates were approximately 50% higher among *An. funestus* (1.75%) compared to *An. arabiensis* (1.16%) and *An. gambiae s.s.* (1.12%), with sporozoite rates highest for *An. funestus* across all 3 geographic regions (Table 3). However, given the relatively low number of sporozoite infected mosquitoes identified, the study lacked precision to evaluate temporal trends in the proportion of mosquitoes infected with sporozoites.

## Insecticide resistance phenotyping

Time response curves for *An. funestus s.l.* exposed to clothianidin and pirimiphos-methyl (active ingredient in Actellic) showed high resistance to clothianidin, but full susceptibility to pirimiphos-methyl (Fig 5). *An. gambiae s.l.* showed susceptibility to clothianidin even at reduced exposure times but mild tolerance to pirimiphos-methyl when exposure time was decreased.

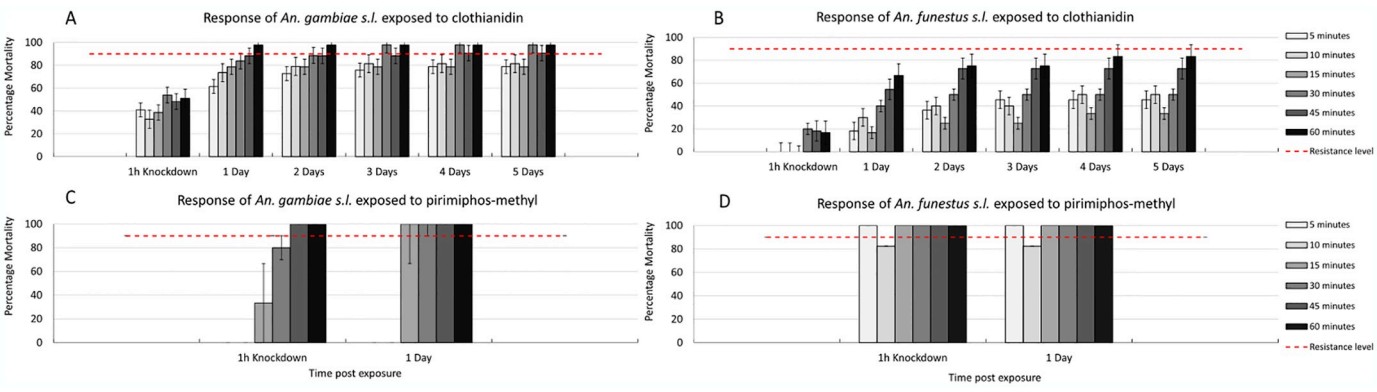

**Fig 5. Mortality** *of An. gambiae s.l.* and *An. funestus* **exposed to clothianidin and pirimiphos-methyl for varying durations.**

## Discussion

The combination of repeated universal LLIN distribution campaigns and sustained IRS has been associated with marked reductions in the burden of malaria at several historically holoendemic areas of Uganda, including Tororo District [15]. However, in 2020–21 malaria began to resurge in these areas following a change in IRS from an organophosphate to clothianidin-based formulations [6]. In this study, we quantified this resurgence and investigated potential causes in a cohort of residents of Tororo District, where IRS had been sustained since 2015, and neighboring Busia District, where IRS has never been implemented. In Tororo, resurgence of malaria increased approximately 18 months after the change to annual rounds of clothianidin-based IRS formulations, reaching a malaria incidence in children of over 4 episodes as compared to 0.5 episodes per year before the change, with over 75% of the population infected with malaria parasites. This resurgence was temporally correlated the emergence of *An. funestus* as the predominant malaria vector. In addition, *An. funestus* had a higher rate of infection with sporozoites compared to the other common malaria vectors and it exhibited high tolerance to clothianidin. Following a shift back to IRS with Actellic, the burden of malaria and the density of *An. funestus* declined sharply in Tororo. In contrast, in Busia district the burden of malaria remained high and relatively stable throughout this period, with *An. gambiae s.s.* the primary vector.

This study and others provide several lines of evidence based on Bradford Hill criteria to support a causal link between the change from an organophosphate (Actellic) to clothianidin-based (Fludora Fusion/SumiShield) IRS formulations and the marked resurgence of malaria. First, the timing of the resurgence (temporality) corresponded to a shift in active ingredient from pirimiphos-methyl to clothianidin-based formulations similar to what has been documented in 4 other districts of Uganda [6]. Second, the timing of the resurgence also corresponded with the emergence of *An. funestus*, which was found to have high tolerance to clothianidin and relatively high sporozoite rates, which likely contributed to a marked increase in transmission intensity (biologic plausibility). Third, the burden of malaria and density of *An. funestus* were sharply reduced when the formulation of IRS was shifted back to Actellic, a formulation for which *An. funestus* was found to be fully susceptible (strength of the association). Finally, just a few kilometers away in the neighboring district of Busia, malaria burden remained relatively unchanged, and the predominant vector was *An. gambiae s.s.*, consistent with historical reports from high transmission areas of Uganda (specificity) [16].

Clothianidin is a neonicotinoid insecticide, one of 4 chemical classes prequalified for use in IRS by the WHO. The two most widely used clothianidin-based IRS formulations have been

SumiShield™ (active ingredient clothianidin alone) and Fludora Fusion™ (active ingredients clothianidin combined with deltamethrin, a pyrethroid), which were prequalified by the WHO in 2017 and 2018, respectively. In preparation for and following the subsequent rollout of clothianidin-based IRS formulations in sub-Saharan Africa, several field-based studies demonstrated broad susceptibility of major Anopheles vector species to clothianidin, with residual activity lasting up to 18 months [17]. In a study of wild pyrethroid resistant *An. gambiae s.l.* carried out in experimental huts in Benin from 2013–14, clothianidin caused high overall mortality rates across a range of wall surfaces [18]. In studies of insectary reared and wild collected mosquitoes conducted from 2016–17 across Africa, *An. gambiae s.l.* from 14 countries and *An. funestus* from 2 countries were widely susceptible to clothianidin [19]. Subsequent studies of field populations of common *Anopheles* species from Kenya, Benin, and Cote d'Ivoire also reported high mortality rates following exposure to clothianidin [20–23]. It should be noted that clothianidin is slower acting than other commonly used insecticide classes, requiring a longer holding period to fully assess post-exposure mortality [18, 24, 25]. Despite these encouraging reports, some recent data have suggested the existence of reduced susceptibility to clothianidin among African vectors, as reported in this study. In a rural area of Cameroon where neonicotinoids have been used for crop protection, field caught *An. gambiae* showed resistance to clothianidin [26]. In a study of laboratory susceptible and resistant laboratory strains using a variety of wall surfaces, the potency of clothianidin was lower against *An. funestus* compared to *An. gambiae* [17]. In a study from Cameroon, Malawi, Ghana, and Uganda, field collected *An. funestus* were broadly susceptible to clothianidin, however, possible cross-resistance was detected in mosquitoes with genetic mutations associated with metabolic resistance [27]. Importantly, in the only other study of the clinical impact of clothianidin-based IRS formulations, from Northern Zambia (where *An. funestus* is the primary vector) in 2019–20, IRS with Fludora Fusion had no impact on parasite prevalence, demonstrating, as in the current study, a lack of clinically relevant efficacy of the insecticide [28].

One of the most striking findings in this and previous studies conducted in Tororo District was the dramatic shift in *Anopheles* species predominance following the implementation of different formulations of IRS. Prior to the implementation of IRS, the primary vector was *An. gambiae s.s.* (74% of female *Anopheles* collected), followed by *An. arabiensis* (22%) and *An. funestus* (4%). After 5 years of IRS with Bendiocarb followed by Actellic, vector densities declined dramatically with *An. arabiensis* making up 99% of female *Anopheles* collected [29]. Entomologic surveillance was not conducted in the area from November 2019 –August 2020, corresponding to the period when IRS with Fludora Fusion was first implemented. However, just prior to the second round of Fludora Fusion, *An. arabiensis* remained the primary vector in houses away from the border in Tororo with a mix of *An. arabiensis* and *An. gambiae s.s.* predominating in houses near the border in Tororo and *An. gambiae s.s.* predominating in in houses on the Busia side of the border. Although low levels of *An. funestus* were detected at baseline in all 3 geographic regions in this study, it was only just before the second round of clothianidin-based IRS that this species emerged as the predominant vector in Tororo, corresponding to the dramatic resurgence of malaria. Interestingly, in houses away from the border in Tororo, *An. arabiensis* again became the predominant species after IRS was shifted back to Actellic, consistent with the trend previously seen when Actellic was used.

*An. funestus* has emerged as an increasingly important malaria vector in many parts of sub-Saharan Africa. In a systematic review of studies from east and southern Africa, the primary vector shifted from *An. gambiae s.l.* between 2000 and 2010 to *An. funestus* between 2011 and 2021 [30]. In addition, several studies have reported higher infection rates among *An. funestus* relative to other *Anopheles* species [30–32]. *An. funestus* has rapidly developed insecticide resistance in many parts of Africa [33–37], is highly anthropophilic, can survive longer than

other vectors [38], and may have the ability to take multiple blood meals within each gono-trophic cycle [39], all factors that may have contributed to the increasing importance of this vector following the scale up vector control throughout Africa.

Our prospective cohort study design allowed for a detailed description of the dynamics and magnitude of clinically relevant measures of the malaria resurgence. The rapid increase in the incidence of symptomatic malaria over a 6–9 month period was similar to historically high transmission areas of Uganda where IRS had been discontinued [15, 40], suggesting that after the 2nd round of Fludora Fusion there was little or no impact, even in a setting of high reported utilization of LLINs. In addition, the peak incidence of malaria during the resurgence exceeded that in children enrolled in another cohort study prior to the implementation of IRS in Tororo District, an observation that was also documented using health facility-based data covering the period from 2011 through 2023 presented in Fig 1 of this study. Not only did the resurgence affect children, but malaria incidence also reached relatively high levels in adults. These findings suggest that the resurgence may have been compounded by a relative loss of naturally acquired immunity to malaria following an extended period when malaria had been well controlled. Despite this suggestion of loss of immunity, there was no evidence from our cohort that there was a "excess" in the risk of severe forms of malaria, highlighting the importance of prompt and effective treatment available 7 days a week at our study clinic. However, a nearby hospital based study conducted during the malaria resurgence reported severe malaria affecting older children with unusual presentations including high risks of prostration, jaundice, severe anemia, and black water fever [41].

This study was not without limitations. First, we used an observational study design, with measures of impact based on comparisons of temporal trends between districts; a cluster randomized controlled trial would have better compared trends, but available resources did not allow this design. Second, there may have been other contributing factors to the resurgence that we did not fully explore, including the dynamics of host immunity. Third, we may have underestimated the overall burden of disease in the community during periods of resurgence, since our cohorts receive prompt treatment and close follow up. Fourth, entomological data were limited to CDC light traps, which do not provide direct measures of biting rates and information on the timing and location of mosquito-to-human transmission. Fifth, measures of insecticide susceptibility did not utilize mosquitoes collected in the study districts; although they were from a nearby district, susceptibility may have differed from that in Tororo District. Despite these limitations, this study benefited from the availability of prior comprehensive surveillance data collected from the area over an extended period, resulting in important scientific findings with significant policy implications.

## Conclusion

Our findings show that resurgence of malaria in Tororo was temporally and geographically correlated with a change in IRS formulation from Actellic to Fludora Fusion and the unique emergence of *An. funestus* as the predominant vector. This unprecedented increase in malaria burden despite sustained and intense vector control combining repeated rounds of universal LLIN distribution and IRS underscores the fragile nature of malaria control in Africa. Rotating IRS formulations has been promoted as a way to limit the selection of insecticide resistance, but in this case such a rotation was associated with a dramatic malaria resurgence in Uganda. Future changes in insecticides should take into consideration on-going surveillance that is comprehensive and includes clinical and entomological metrics, including vector species composition and insecticide resistance patterns. Surveillance data should also be timely and actionable, allowing for well-coordinated responses by policy makers. Indeed, the WHO Global

Malaria Programme now encourages the use of multiple sources of data to better understand malaria risk at the sub-national level and to target interventions as part of its "High Burden High Impact" strategy [42].

## Acknowledgments

We thank all the study team members for successfully conducting the PRISM studies over the years and the Tororo and Busia district administrations for their support. We are grateful to the study participants who participated in this study and their families.

## Author Contributions

**Conceptualization:** Moses R. Kamya, Joaniter I. Nankabirwa, Emmanuel Arinaitwe, Grant Dorsey.

**Data curation:** John Rek, Maato Zedi, Catherine Maiteki-Sebuguzi, Ambrose Oruni, Bryan Greenhouse, Jessica Briggs, Paul J. Krezanoski, Peter Olwoch, Grant Dorsey.

**Formal analysis:** Grant Dorsey.

**Funding acquisition:** Moses R. Kamya, Joaniter I. Nankabirwa, Grant Dorsey.

**Methodology:** Moses R. Kamya, Joaniter I. Nankabirwa, John Rek, Catherine Maiteki-Sebuguzi, Jimmy Opigo, Sarah G. Staedke, Ambrose Oruni, Martin J. Donnelly, Bryan Greenhouse, Jessica Briggs, Teun Bousema, Philip J. Rosenthal, Peter Olwoch, Prasanna Jagannathan, Grant Dorsey.

**Supervision:** Moses R. Kamya, Joaniter I. Nankabirwa, Emmanuel Arinaitwe, Catherine Maiteki-Sebuguzi, Jimmy Opigo, Sarah G. Staedke, Martin J. Donnelly, Paul J. Krezanoski, Teun Bousema, Philip J. Rosenthal, Prasanna Jagannathan, Isabel Rodriguez-Barraquer, Grant Dorsey.

**Writing – original draft:** Moses R. Kamya, Joaniter I. Nankabirwa, Emmanuel Arinaitwe, Ambrose Oruni, Grant Dorsey.

**Writing – review & editing:** Moses R. Kamya, Joaniter I. Nankabirwa, Emmanuel Arinaitwe, John Rek, Maato Zedi, Catherine Maiteki-Sebuguzi, Jimmy Opigo, Sarah G. Staedke, Ambrose Oruni, Martin J. Donnelly, Bryan Greenhouse, Jessica Briggs, Paul J. Krezanoski, Teun Bousema, Philip J. Rosenthal, Peter Olwoch, Prasanna Jagannathan, Isabel Rodriguez-Barraquer, Grant Dorsey.

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
