## [Decision Letter · Decision Letter 0]

24 Apr 2024

PGPH-D-24-00571

Dramatic resurgence of malaria after 7 years of intensive vector control interventions in Eastern Uganda

Dear Dr. Kamya,

Thank you for submitting your manuscript to PLOS Global Public Health. After careful consideration, we feel that it has merit but does not fully meet PLOS Global Public Health’s publication criteria as it currently stands. Therefore, we invite you to submit a revised version of the manuscript that addresses the points raised during the review process.

We look forward to receiving your revised manuscript.

Kind regards,

Ruth Ashton, Ph.D.

Academic Editor

Journal Requirements:

1. Please provide separate figure files in .tif or .eps format only and remove any figures embedded in your manuscript file. Please also ensure all files are under our size limit of 10MB.

Additional Editor Comments (if provided):

Reviewers' comments:

Reviewer's Responses to Questions

**Comments to the Author**

1. Does this manuscript meet PLOS Global Public Health’s publication criteria? Is the manuscript technically sound, and do the data support the conclusions? The manuscript must describe methodologically and ethically rigorous research with conclusions that are appropriately drawn based on the data presented.

Reviewer #1: Yes

Reviewer #2: Partly

2. Has the statistical analysis been performed appropriately and rigorously?

Reviewer #1: Yes

Reviewer #2: Yes

3. Have the authors made all data underlying the findings in their manuscript fully available (please refer to the Data Availability Statement at the start of the manuscript PDF file)?

Reviewer #1: Yes

Reviewer #2: Yes

4. Is the manuscript presented in an intelligible fashion and written in standard English?

Reviewer #1: Yes

Reviewer #2: Yes

5. Review Comments to the Author

Reviewer #1: The manuscript is well written and presents well the landscape of the study areas. Such observational studies are important for countries to be able to make evidence-based decisions beyond trials. The countries have routine data and it's important to make good use of them as the study did, in addition to the cohort for collecting additional data. I think the findings. I understand the fact that one study and manuscript cannot include/cover all the factors that related to malaria burden. It would of value to add some discussions around any potential variations (or not) in rainfall and temperatures that could also have an influence in entomological variables. Similarly, I don't think I saw any information about the transmissions seasons in the study areas. It's possible that there is no clear high transmission period(s) in the neither districts - but still good information for context. The limitations are spot on - would have benefitted from additional entomological collection/data to really appreciate the potential changes in vector behaviors. The resurgence is quite impressive - in addition to the potential changes in host immunity, I wonder about the potential changes in human behavior - in a community that have benefited from IRS (with a vector susceptible to the insecticide).

A couple of small (preferential) suggestions - Consider rephrasing lines 340-344, I think the "each of the first 7 months of observation" should be replaced with simply "the first 7 months of observations".

In a few places, you introduce description with "briefly" - I don't think it necessary.

Reviewer #2: The paper on Dramatic resurgence of malaria after 7years of intensive vector control interventions in Eastern Uganda presents interesting results. However, there several concerns that may need addressing before it may be fit for publication.

Comments:

Line # 28-29: Clearly from your results, IRS with carbamates & Actellic was implemented during the 5years of dramatic decrease. As such, the use of the term “following 5 years of…” may be misleading as it suggests that IRS was implemented for 5years after which, there was a dramatic decrease. Please consider using a more appropriate term such “during 5 years …” to avoid confusion.

Line # 45: It may not be clear to the reader of the abstract when you report average monthly cases without a specific period covering the months for which the average was generated.

Line # 57: You summarize sporozoite rate simply as approximately 50%, again without indicating the period being considered, given that you define distinct durations for consideration in the paper. One would have expected sporozoite results to be presented by the durations under consideration.

There are many punctuation issues in the paper that may need to be cleaned up. For instance,

Line # 70: Between 2000 and 2015, it was estimated that …

Line # 80: … has been slow and difficult to sustain, …

Line# 129: As in Tororo prior to 2013, vector …

Line # 187-189: … clinic visit, …

And many, many other places that punctuation needs to be improved.

Importantly also, your punctuation around in-text citations needs consideration. It’s rather strange that your punctuation comes before your in-text citations. Please double check PGPH recommended on citation. This also goes for the use of “(“ rather than “[“ as recommended by the PGPH journal.

There is not a clear section in the paper, either in the methods, results or discussions that addresses the part of the study aim pertaining to “causes of resurgence”. This may need to be clarified.

Line # 90-91: This sentence may need to be simplified to avoid confusing the reader on what happened when. For instance, stating this as “Comparing key malaria indicators measure prior to IRS (2011-14 and after two rounds of LLIN as well as 5 years of sustained IRS (2017-19), …” or simpler, would help the reader know when there was or the wasn’t IRS and/or LLIN.

Line # 94: The reported “marked shift in predominant vector” may need a reference. It’s also not clear whether this marked shift was all through 2011-19 or that you’re referring to 2017-19.

Line #118-134: Lots of information is provided here without a single reference. Certainly, one would not expect that all these LLIN and IRS activities were part of your study or its data collection in the field. Please provide appropriate sources of this valuable information.

Line # 130: The sentence may need to be corrected. “Universal distribution campaigns were … “

Line #155-158: It would be helpful to provide a brief rationale for the seemingly dissimilar definition of near-border households between the two districts. In one, it was 1.9 – 3.2Km and in the other 0.7-3.5Km areas

It’s also not quite clear why in Busia, there was no category of households characterized as “away from the border”. Was the general assumption that every where outside of IRS cover would be the same? This would be strange, given that you indicated that your plan was to examine the gradient over this transmission area. If not, this may be a limitation to your capacity to fully understand the gradient of interest.

Line # 166: From an ethical stand point, were the 4 households that were replaced included in any of the analyses, be it even for the duration prior to their withdrawal? This may need to be made clear because, they are not expected to be included at all, given their withdrawal of consent.

Line #281: As seen from your Figure 3 A, episodes / person years were already on an upward trend. Given this, how did you define “increase accelerated” or are you referring to the increase not having been slowed or disrupted? This terminology may be misleading.

Line # 284-285: As your Figure 3A clearly shows, malaria decline doesn’t appear to be tied to actellic alone. From your results, even the location with no IRS showed the same level of decline of malaria burden. You may want to tone this claim down to be more consistent with your results.

Line #315: It’s not clear or indeed confusing why after indicating that the duration of assessment of incidence as between Apr-2021 – Sep-2023 in the text before this sentence, you then start describing this part from Feb-Aug 2021, which included a large chunk of a duration outside of the stated study duration.

Line #387: Please clarify why since you set out to evaluate causes of resurgence including entomological measures as indicated in your methods, that then you lacked the precision to evaluate temporal trends in proportions of mosquitoes infected with sporozoites.

Line #517-518: It’s also surprising that your study {that was “designed to characterize the resurgence and explore potential causes…”, as stated in prior to this, did all the entomological procedures extensively explained and later elaborated on in the discussion} reports the limitation of susceptibility that did not utilize mosquitoes in the study districts.

You may consider revisiting the strong worded aim of the study to this effect. Otherwise, it appears that this study was designed to assess something else and possibly not exactly the resurgence. If not, then this limitation is not expected.

Line # 524: As it is, the conclusion drawn around resurgence being geographically corelated with change in IRS formulation is not supported by your results. For instance, 1) Your results indicated a fairly similar temporal trend in burden (episodes per 1000-person years) across site-geographies. 2) You didn’t have similar strata in Busia as you did in Tororo (near versus away from border) to adequately examine geographical variations.

Line # 525: Stating that resurgence was “correlated” with change in IRS may also be an overstatement as clearly shown in your results. Importantly, FF IRS occurred in the middle of a heavily developing burden peak. It’s would not be true that the peak was accelerated following this IRS change.

One would argue, rather strongly and also consistent to what you stated elsewhere in the paper and supported by your Figure 3A, that there is an apparent slower action of FF, which delays and/or dampens the slope after the peak occurring during the FF spray round.

Line # 533-534: Given your indication that your program has been conducting surveillance in this same site since 2011. Are you, by recommending that surveillance data should be timely…” implying that your surveillance data was not timely or available to the program to foster a well-coordinated implementation of this round of IRS.

Rather than this rather distal conclusion, one would have expected some aspects of the conclusion to address the part of the aims of your study concerning the causes of this resurgence, which is currently missing.

Figure 1: The legend refers to PRISM1 and 2 study sites yet there seems to be only one site (purple) on the map.

Figure 4: Your use of the same exact colors that represent sites in other Figures for vector types may need to be revisited to enable readers better distinguish these results.

Please consider using the same y-axis scale in this Figure, especially for A and B, as they are both referring to Tororo

Figure 5: The x-axis label of “Days post exposure” seems inappropriate unless you intend to represent 1hr in terms of days. It could be better stated as “time post exposure”.

Author contributions: It appears that there are so many co-authors without specified contributions to the paper. Say for instance, Jimmy Opigo, Teun Bousema, Phil Rosenthal, … and many others.

Table 3: There are several N quantities presented that are not fully explained. For instance, there’s N* in the column heading, then *N in the footnote, and then N when describing VD. These may need to be clarified clearly spelling out which is which.

In a number of your tables, why is there inconsistent use of 2 and in other cases 1 decimal places, especially Tables 2 and 3? Please consider making this consistent.

6. PLOS authors have the option to publish the peer review history of their article (what does this mean?). If published, this will include your full peer review and any attached files.

**Do you want your identity to be public for this peer review?** For information about this choice, including consent withdrawal, please see our Privacy Policy.

Reviewer #1: No

Reviewer #2: No

---

## [Editor Report · Decision Letter 1]

16 May 2024

PGPH-D-24-00571R1

Dramatic resurgence of malaria after 7 years of intensive vector control interventions in Eastern Uganda

Dear Dr. Kamya,

Thank you for submitting your manuscript to PLOS Global Public Health. After careful consideration, we feel that it has merit but does not fully meet PLOS Global Public Health’s publication criteria as it currently stands. Therefore, we invite you to submit a revised version of the manuscript that addresses the points raised during the review process.

We look forward to receiving your revised manuscript.

Kind regards,

Ruth Ashton, Ph.D.

Academic Editor

Journal Requirements:

2. Please ensure that Funding Information and Financial Disclosure Statement are matched.

3. In the Funding Information you indicated that no funding was received. Please revise the Funding Information field to reflect funding received.

4. Some material included in your submission may be copyrighted. According to PLOS’s copyright policy, authors who use figures or other material (e.g., graphics, clipart, maps) from another author or copyright holder must demonstrate or obtain permission to publish this material under the Creative Commons Attribution 4.0 International (CC BY 4.0) License used by PLOS journals. Please closely review the details of PLOS’s copyright requirements here: PLOS Licenses and Copyright. If you need to request permissions from a copyright holder, you may use PLOS's Copyright Content Permission form.

Potential Copyright Issues:

Fig 1: please (a) provide a direct link to the base layer of the map (i.e., the country or region border shape) and ensure this is also included in the figure legend; and (b) provide a link to the terms of use / license information for the base layer image or shapefile. We cannot publish proprietary or copyrighted maps (e.g. Google Maps, Mapquest) and the terms of use for your map base layer must be compatible with our CC-BY 4.0 license. 

Additional Editor Comments (if provided):

Thank you for this revision. While almost all peer reviewer feedback has been addressed, I would like to request further clarifications on the following:

- Question 16 from reviewer #2 was not fully addressed, and I agree that it is a little confusing to mention that incidence was analysed in 6-month chunks, then in the next sentence to describe incidence in a way that does not use the same 6-month breakdown. Perhaps a clarification is needed at the start of the sentence to emphasize that you are now assessing the monthly incidence estimates.

- I suggest to adjust the colour schemes for IRS types used in figure 2 to match those in figures 3 & 4. In addition, is there a particular reason why in figure 2 the whole period is shaded to match specific IRS spray schemes, while in figure 3 & 4 only the period when IRS campaign occurred is shaded? If the intention is to indicate the period of expected residual effectiveness in figure 2, please clarify this in the figure description.

- Regarding the response to question 20 from reviewer 2, you stated that you took care not to imply that you had proven a causal hypothesis, however in line 428 you state that the evidence presented does support a causal link. If you do wish to indicate that these evidence are indicative of a causal link between changing IRS chemicals and resurgence, I recommend referring to or framing this around the Bradford-Hill criteria or other causal inference literature using observational study designs.
---

## [Editor Report · Decision Letter 2]

11 Jun 2024

Dramatic resurgence of malaria after 7 years of intensive vector control interventions in Eastern Uganda

PGPH-D-24-00571R2

Dear Dr. Kamya,

We are pleased to inform you that your manuscript 'Dramatic resurgence of malaria after 7 years of intensive vector control interventions in Eastern Uganda' has been provisionally accepted for publication in PLOS Global Public Health.

Best regards,

Ruth Ashton, Ph.D.

Academic Editor